# Sparse Logistic Regression-Based EEG Channel Optimization Algorithm for Improved Universality across Participants

**DOI:** 10.3390/bioengineering10060664

**Published:** 2023-05-31

**Authors:** Yuxi Shi, Yuanhao Li, Yasuharu Koike

**Affiliations:** 1School of Engineering, Tokyo Institute of Technology, Yokohama 226-8503, Japan; li.y.ay@m.titech.ac.jp; 2Institute of Innovative Research, Tokyo Institute of Technology, Yokohama 226-8503, Japan; koike@pi.titech.ac.jp

**Keywords:** electroencephalogram, channel optimization, sparse logistic regression, brain–computer interface

## Abstract

Electroencephalogram (EEG) channel optimization can reduce redundant information and improve EEG decoding accuracy by selecting the most informative channels. This article aims to investigate the universality regarding EEG channel optimization in terms of how well the selected EEG channels can be generalized to different participants. In particular, this study proposes a sparse logistic regression (SLR)-based EEG channel optimization algorithm using a non-zero model parameter ranking method. The proposed channel optimization algorithm was evaluated in both individual analysis and group analysis using the raw EEG data, compared with the conventional channel selection method based on the correlation coefficients (CCS). The experimental results demonstrate that the SLR-based EEG channel optimization algorithm not only filters out most redundant channels (filters 75–96.9% of channels) with a 1.65–5.1% increase in decoding accuracy, but it can also achieve a satisfactory level of decoding accuracy in the group analysis by employing only a few (2–15) common EEG electrodes, even for different participants. The proposed channel optimization algorithm can realize better universality for EEG decoding, which can reduce the burden of EEG data acquisition and enhance the real-world application of EEG-based brain–computer interface (BCI).

## 1. Introduction

Brain–computer interface (BCI) is a system that can convert brain wave signals into external control commands [1,2]. BCI can be classified in a variety of ways, including invasive and non-invasive BCI [3,4], passive and active BCI [5,6], and so on. Its six main application scenarios include: research, improve, enhance, restore, replace, and supplement [7]. For rehabilitation, the recent trends in BCI mainly consider motor imagery (MI) and transcranial direct current stimulation applied to target brain areas, which can induce neural plasticity in chronic stroke patients’ white matter and cortical function [8]. BCI-driven brain region magnetic stimulation has been proven to increase cortical activation in stroke patients [9]. BCI is also used to detect human emotions and explore cognitive intent decoding of senders [10], converting it into commands for the recipient’s brain in brain-to-brain interface (BBI) experiments [11,12,13]. Additional applications of BCI include brain fingerprinting for lie detection [6], detecting drowsiness to improve human work performance [14,15], estimating reaction time [16], controlling virtual reality [17], quadcopters [18], and video games [19], as well as driving humanoid robots [20,21]. On the other hand, online BCI systems normally focus on computational cost, equipment cost, and classification accuracy [22]. Excessive electrodes will increase the setting, maintenance, and calculation costs. To address this problem, electroencephalograph (EEG) channel selection accomplished by removing redundant electrodes while maintaining or even improving BCI classification accuracy is an effective technique [23].

The main channel selection techniques are as follows: filtering techniques that use autonomous assessment criteria contain the methods of mutual information [24] and chi-squared [25]; wrapper techniques assess a subset by employing a predictor and its resulting output as the objective function, including the sequential selection algorithm [26,27] and heuristic search algorithm [28]; hybrid techniques that combine the filtering and wrapper techniques [29]; embedded techniques that depend on the learning process of a specific classifier to create standards, which contain recursive feature elimination for support vector machines (SVM-RFE) [30] and feature selection–perceptron (FS-P) [31]. In particular, for the MI task, numerous EEG channel selection methods have been proposed to reduce the computational complexity, such as CSP-based filtering techniques (CSP variance maximization methods) [32], methods based on CSP variants [33], non-CSP-based filtering techniques (information measure-based methods) [34], and so on. Furthermore, the channel selection based on correlation coefficients (CCS) through Pearson correlation analysis of EEG data between each pair of channels is also a widely used filtering technology [35,36,37].

Because the brain activity of each participant is independent, most channel optimization algorithms show differences in the selected electrodes when applied for individual participants [38,39]. The improvement of the universality for EEG channel optimization, referring to how effectively the selected channels can be generalized to different participants, has not received a great deal of attention in the literature. The hypothesis of this research is that by analyzing the channels that contain more features utilized in the group analysis and by choosing the frequently involved channels as the common channels, the universality of channel optimization algorithm can be improved for the group analysis.

Sparse logistic regression (SLR) [40] is a sparse classification model that can realize classifier training and feature selection simultaneously. Due to its parameter-free property and robustness against over-fitting, SLR has been widely applied for brain activity analysis, including EEG [41,42,43,44], fMRI [40,45,46,47], and cortical current source [48,49]. Inspired by the generalization performance of the effective classification method SLR [50], this study proposes an SLR-based EEG channel optimization algorithm. The key motivation is that SLR will generate sparse feature weight vectors after the model training process. By analyzing the number of nonzero weights for each channel, the subsequent channel optimization can be realized. The proposed algorithm was evaluated for both the individual and the group level, compared with the conventional CCS-based selection method regarding universality. In addition, most existing algorithms attempt to improve selection effectiveness by finely processing data. However, EEG signals in real-time applications are usually contaminated with noise and artifacts. Therefore, this paper verifies the algorithm’s effectiveness using raw data. The confusion matrix is usually used to represent classification results [51] and, in this research, the proportion of correctly classified samples in the confusion matrix is utilized to indicate decoding accuracy for evaluating channel optimization. In general, this study presents a straightforward and effective method for optimizing channels in BCI, which can be applied both individually and in a group, facilitating low-cost implementation.

The structure of this paper includes the following: Section 2 describes the utilized datasets, the proposed algorithm, CCS used for comparison, and evaluation procedures. Section 3 presents the performance of the proposed algorithm for individuals and groups, as well as the performance of the CCS algorithm for groups. Section 4 focuses on evaluating the proposed algorithm in terms of its performance, universality, robustness, and comparison with CCS, etc. Finally, the research in this paper is briefly concluded in Section 5.

The highlights and contributions of this research are as follows:1.A straightforward and universal algorithm is proposed to optimize EEG channels for improved universality across participants.2.The raw data are used to verify the effectiveness of the proposed algorithm in both individuals and populations.3.This method is helpful in significantly reducing redundant channels, solving the problem of channel selection diversity, and contributing to the development of low-cost BCI.

## 2. Materials and Methods

### 2.1. Datasets

#### 2.1.1. Main Dataset

The MI brainwave dataset based on prediction errors was used for verification [43]. It included data from 10 healthy participants (8 males, 2 females, aged 21–25 years). The EEG data were recorded from 64 active channels (sampling rate: 512 Hz), and a custom-designed Galvanic Vestibular Stimulation (GVS) device was utilized to elicit prediction errors during participants’ MI, generating a 1 Hz sine wave lasting for 0.5 s with a maximum intensity of 0.4 mA. Each participant performed two MI tasks (left/right side and front/back side), with six sessions per task and 60 trials per session (for a total of 360 trials). In each trial, the voice cue was presented initially followed by a 3 s delay before applying GVS for a duration of 0.5 s. Participants were instructed to perform MI tasks upon hearing the voice cues until entering a 3 s rest period. Since the prediction errors induced by 0.5 s GVS can achieve satisfactory decoding performance suitable for future applications, data from this period were selected to verify the proposed algorithm.

#### 2.1.2. Supplemental Dataset

In the aforementioned experiments, the time period of approximately 3 s surrounding pure MI was extracted as a supplementary dataset. In the left/right MI task and front/back MI task, each direction contained 180 trials. The dataset was also used to evaluate the proposed algorithm in the individuals and in the group.

### 2.2. Proposed SLR-Based EEG Channel Optimization Algorithm

Raw data in close proximity to their original state were used for analysis in this study. For each participant, five periods of 0.5 s GVS time (each 0.1 s, 512 Hz, corresponding to 51 samples) were analyzed from their datasets. Within each time period, the participant completed a total of 360 trials that included EEG data and label value information. These trials were randomly divided into an 80% training set and a 20% testing set before being inputted into a MATLAB function tool [40] for analysis (this process was repeated 20 times). As a result, a weight vector was obtained for each participant that was constructed as [(64 channels × 51 samples) × 20 repetitions × 5 time periods] and was saved for subsequent channel optimization.

#### 2.2.1. Channel Optimization

The nonzero absolute elements in the weight vector of each participant were sorted in descending order. The interval of full/top 50%/top 25% of these sorted elements along the channel dimension was calculated, and the count value for each channel was utilized as a basis for optimization.

First, we attempted to carry out this operation in individuals, called individual analysis. The algorithm is shown in Figure 1 and is described as follows (Algorithm 1).

The optimization algorithm described above was analyzed on a participant-by-participant basis (individual analysis). However, since the channels optimized for different participants vary, this approach is not conducive to practical application and cost reduction. Therefore, a group-based channel optimization method for participants is also planned.
**Algorithm 1** Individual analysis**input**: individual weight vector of participants**output**: the ranked channels of participants separately1.The initial weight vectors are obtained as weight vector ‘W’;2.The nonzero absolute value elements in the weight vector ‘W’ are arranged in descending order, and the interval of top 50%/top 25% is defined;3.In the weight vector ‘W’, the number of absolute elements that fall within the 3 intervals (full/top 50%/top 25% intervals) are counted respectively along the channel dimension, and the weight count vector can be recognized as [counting values of 64 channels × 3 kinds of intervals];4.The weight count vector in step (3) is ranked in descending order, and the ranked channels of three counting intervals can be acquired, structured as [64 descending ordered channels × 3 kinds of intervals];5.The descending ordered channels of three different counting intervals are evaluated.

In group analysis, the weight vectors of all participants are analyzed as a single unit, and then the channels of the participant group are optimized according to the steps shown in the related phase of Figure 1 (Algorithm 2).
**Algorithm 2** Group analysis**input**: weight vector of participant group**output**: the ranked channels of the participant group1.The initial weight vectors of the group (all 10 individuals) are obtained and reconstructed as the group weight vector ‘W’;2.The nonzero absolute value elements in the group weight vector ‘W’ are arranged in descending order, and the interval of top 50%/top 25% is defined;3.In the weight vector ‘W’, the number of absolute elements that fall within the 3 intervals (full/top 50%/top 25% intervals) are counted respectively along the channel dimension, and the weight count vector can be recognized as [counting values of 64 channels × 3 kinds of intervals];4.The group weight count vector in step (3) is ranked in descending order, and the ranked channels of 3 counting intervals can be acquired, structured as [64 descending ordered channels × 3 kinds of intervals];5.The descending ordered channels of three different counting intervals are evaluated.

#### 2.2.2. Evaluation of Channel Optimization

Referring to the bottom right corner of Figure 1, after applying the optimization algorithm to analyze the channels, a sequence of channels was obtained. Starting from the most relevant channel (i.e., the top-ranked channel), each channel was added to the decoding queue based on the SLR-VAR classification algorithm and analyzed one by one. Eventually, an optimal decoding performance queue of channels could be determined for subsequent comparative analysis.

After the SLR-based EEG channel optimization algorithm was verified in individuals and the participant group (left/right MI task and front/back MI task), the performance was compared in Figure 2. In addition, the brain distribution map of weight count values for channels in group analysis (the depth of color represents different channel weights) is shown in Figure 3, with commonly optimized channels marked by red circles among individual participants. The decoding performance of these commonly selected channels was compared between the individuals and the participant group, and their universality was evaluated and discussed.

#### 2.2.3. Verification of CCS-Based Channel Selection Method

This research also compared the classical channel selection method CCS with the proposed algorithm in terms of the universality of channel optimization. The CCS method was used to evaluate event-related activations in the brain, which is widely used for individual participant channel screening [35,36]. The mechanism is as follows: channels that contain the same MI task have more common information, resulting in larger correlation coefficients. Conversely, channels not related to the MI have fewer common features and therefore smaller correlation coefficients. The correlation coefficient was calculated using Z-score normalization to reduce time variability in each brain wave dataset. Pearson correlation analysis was used to calculate the correlation coefficient, which quantifies the linear correlation between two or more random variables using the following formula:(1)$$\rho \left(X,Y\right)=\frac{1}{n-1}\sum_{i=1}^{n}\left(\frac{X_{i}-\bar{X}}{\sigma_{X}}\right)\left(\frac{Y_{i}-\bar{Y}}{\sigma_{Y}}\right)$$
where *X* and *Y* are the data of two different channels, *n* is the number of experimental tasks, $\bar{X}$ and $\bar{Y}$ are the mean values of the data, and $\sigma_{X}$ and $\sigma_{Y}$ are the standard deviations.

The flowing calculation process was conducted for the participant group (Algorithm 3):
**Algorithm 3** The analysis of CCS in the participant group**input**: object EEG data of the participants**output**: the ranked channels of the participant group1.0.5 s GVS period data are divided into different groups according to different tasks;2.The correlation coefficient vector of each task is calculated separately;3.The correlation coefficient vector of all the tasks are synthesized, and the mean value of each channel is calculated to obtain the average correlation coefficient vector;4.The average correlation coefficient vector across the participants is synthesized and ranked in descending order, and the common descending ordered channels are obtained;5.According to the descending order, the same evaluation of channel optimization is conducted, and the decoding results are saved for subsequent comparison.

## 3. Results

Firstly, the main dataset was used to validate the performance and universality of the CCS method. The average correlation coefficient for each channel was obtained through pairwise correlation coefficient analysis in the participant group and then sorted accordingly. After the rank of optimized channels was observed, the evaluation was conducted. In the left/right MI task, the mean decoding accuracy of the optimal decoding queue that was analyzed by the CCS method across the participant group was 85.58% (±5.84), and the required number of channels is 56; in the front/back MI task, it is 81.83% (±4.42) and 8 channels (see the black bar and related number in Figure 2). Moreover, the number of channels of the optimal decoding queue in the CCS exceeded that of the algorithm proposed in this research.

Next, the proposed algorithm’s channel optimization was thoroughly evaluated for three counting intervals for both the individuals and for the participant group. The ranked channels were added to the decoding queue one by one in order to obtain decoding results. The optimal decoding accuracy for individual analysis and group analysis, as well as the related number of channels, are shown in Figure 2.

Generally, it can be inferred that the optimal decoding queue typically consists of 2–16 channels, resulting in a reduction of redundant channels by 75–96.9%. When the full interval, the top 25% interval for the left/right MI task, the top 50% interval, and the top 25% interval for the front/back MI task are considered, the SLR-based EEG channel optimization algorithm requires fewer channels in group analysis compared to individual analysis. Meanwhile, the decoding accuracy increased 1.65–5.1% when compared with the decoding results by using all 64 channels.

As shown in Figure 2a, there is no significant statistical difference in the left/right MI task compared to the group analysis, although the optimization algorithm performs better in individual analysis; however, it is significantly better than the CCS method (p<0.01, two-sample *t*-test). In addition, in the front/back MI task, the proposed algorithm outperforms the group analysis, with a statistically significant difference in the full interval and in the top 50% interval (p<0.01, two-sample *t*-test), and the optimization algorithm also outperforms the CCS method (p<0.01, two-sample *t*-test).

Since in the left/right MI task, the proposed algorithm is more likely to achieve better results for the individuals, the channels commonly selected for individual analysis of participants in the left/right MI task and the front/back MI task were also analyzed to further compare the universality in group analysis, and the average decoding accuracy is shown in Table 1. The distribution of weight count values in the group analysis of the proposed algorithm is illustrated in Figure 3. Additionally, channels that were commonly selected among individuals based on their individual analysis are highlighted with red circles (keeping the same number of channels as in the optimal decoding queue from the group analysis).

As can be seen in Table 1, in all three counting intervals of left/right MI task and front/back MI task, the proposed SLR-based EEG channel optimization algorithm demonstrates higher accuracy in the group analysis. Moreover, the highest decoding performance is achieved when weights are counted within the top 25% interval.

## 4. Discussion

### 4.1. Comparison of the SLR-Based EEG Channel Optimization Algorithm and CCS in the Group Analysis

According to the evaluation of channel optimization, there was a slight increase in decoding accuracy as the number of CCS-optimized channels increased in the decoding queue for the left/right MI task. This phenomenon suggests that highly relevant channels for decoding may not be identified and ranked even when selecting the top 56 channels, and the optimal average decoding accuracy is 85.58% (±5.84). Furthermore, even when the proposed SLR-based EEG channel optimization algorithm achieved the best decoding performance of 86.63–87%, the CCS method still yielded results ranging from 82.3% to 84.06% with an equal number of channels in the decoding queue. In the front/back MI task, the proposed algorithm can achieve satisfactory decoding performance with only 3 channels (across the 3 counting intervals), while CCS still requires 8 channels, and its decoding performance is obviously worse than that of the proposed algorithm (p<0.01, two-sample *t*-test).

### 4.2. Performance of the Proposed Algorithm in Supplemental Datasets

In the left/right MI task, optimal decoding accuracy of individual analysis (across 3 counting intervals) is higher than the group analysis (up to 1.38%); in the front/back MI task, the maximum diversity is 1.27%. However, there is no significant difference in analysis between individuals and the participant group (namely, p>0.01, two-sample *t*-test). Therefore, the proposed algorithm can still be used as a universal channel to reduce equipment costs.

### 4.3. Optimization Performance in the Individuals and the Participant Group

Individual differences exist in the analysis of EEG data and related channel selection; therefore, collective universality should be considered for large-scale applications. Results from verifying the main dataset show that the proposed algorithm performed better for left/right MI task among individuals, while it performed better for front/back MI task in the participant group. This means that it is not easy to improve performance in the left/right MI task by counting the weights that fall into the same interval defined by the group. We believe that the counting interval is a fundamental factor that affects the ranking of recognized channels. In the delineation scale of the three counting intervals (full/top 50%/top 25%), only the full interval considers nonzero absolute weights, while the top 50% and the top 25% intervals require predefined object weight vectors for individuals and the participant group (refer to the Algorithms 1 and 2). This difference is also reflected in the absolute weight values of the lower boundary for each interval, represented as the lower boundary values of the interval. The distribution of the lower boundary values for the top 50%/top 25% intervals between the individual analysis and the group analysis is shown in Figure 4. By comparing the upper and lower rows of the graph, it can be concluded that, in the left/right MI task, whether it is the top 50% or top 25% interval, there is a significantly larger discrepancy between the lower boundary values of individual intervals (the bar graphs) and those of the group (the red line), compared to that in the front/back MI task. That is, in the left/right MI task, the mean deviation of the lower boundary values of the top 50%/top 25% intervals between individuals and the group are 7.70 and 16.01, respectively, while in the front/back MI task, they are 3.64 and 7.56. Therefore, in the left/right MI task, the unified interval calculated for the group differs greatly from that of the individuals, which is reflected in the ranking of channels and corresponding optimization performance.

In addition, although the proposed algorithm makes it easier to achieve better performance in individuals during the left/right MI task, the possibility of finding commonly selected channels among individuals was also verified, and the performance was compared with the group analysis. Results show that, although the optimization performance of the proposed algorithm in the group analysis was inferior to that of the individual analysis for the left/right MI, the selected channels were still more representative than those commonly selected among individuals. Meanwhile, there is no significant difference in decoding performance between the group analysis and the individual analysis for the left/right MI task (average diversity was less than 0.46%, p>0.01, two-sample *t*-test). Even at the top 25% interval in the left/right MI task and the top 50%/25% intervals in the front/back MI task, the proposed optimization algorithm required fewer channels for group analysis than for individual analysis.

Furthermore, as depicted in Figure 4, the lower boundary for individual analysis of Participant No. 3 exhibits the greatest discrepancy with that of the group analysis. Therefore, individual analysis may be employed for these participants, while group analysis with a uniform interval can be utilized for all other participants; however, this approach may increase computational costs and reduce generalizability.

### 4.4. Impact of Different Counting Intervals on the Proposed Algorithm in Group Analysis

As shown in Table 1, the highest decoding accuracy was mostly achieved by counting the absolute weights within the top 50%/top 25% intervals (left/right and front/back MI tasks). It can be speculated that higher intervals result in higher decoding-related weights, whereas roughly evaluating full intervals leads to many redundant features being recognized as useful, which affects decoding accuracy. This is consistent with intuitive experience. On the other hand, although the decoding accuracy is slightly decreased, the ‘full interval’ does not require calculation of the predefined value for the lower boundary of the interval and requires fewer channels (10 for the left/right MI task and 3 for the front/back MI task) to achieve optimal decoding performance. Therefore, these three counting intervals can be selected and used in specific circumstances.

### 4.5. Spatial Distribution of the Weight Count Values and Related Neurophysiological Significance

Figure 3 shows the spatial distribution of weight count values in group analysis using the proposed optimization algorithm to analyze the prediction error-based MI dataset (main dataset). The channels commonly selected by individuals (marked with red circles) and those usually selected by the optimization algorithm in the group (darker channels) mostly overlap. In the left/right MI task, when narrowing the counting interval from ‘all’ to the top 25%, dark channels are concentrated in the frontal lobe, particularly on the left anterior side (electrodes designated FP1, AF7, AF3, F7, FT7, FC1, etc.). In the front/back MI task, as the interval is narrowed, commonly recognized channels are concentrated in the frontal lobe and towards the right frontal lobe (electrodes Fpz, AFz, AF4, F6, F8, FC4, etc.). By comparing Figure 3(a1–a3,b1–b3) in terms of the 3 counting intervals, it can be observed that narrowing the counting interval results in a decrease in dark channels and an increase in redundant channel identification.

The channels distributed in the primary sensory motor areas, posterior parietal cortex, and cerebellum correspond with the activation of prediction errors [52]. These dark channels located in the frontal lobe play an important role in autonomous behavior, including the motor cortex responsible for actions such as walking. The frontal lobe also plays a role in judging the consequences of current behaviors. In the main dataset experiment, participants received subconscious stimuli after making MI. The participants’ brains kept updating the forward model, which is largely related to the mechanism of prediction errors. Some other darker channels are located in the parietal lobe, which is thought to be important for processing somatosensory input [53] as it receives information from subconscious stimuli.

### 4.6. Robustness of Proposed Algorithm

The EEG signal is a non-stationary signal that contains abundant noise and artifacts [54]. Unlike recent studies on channel selection algorithms, this research used raw EEG data (which was only downsampled) from the original data for verification. This approach is closer to online application situations and reduces preprocessing and feature selection time. Furthermore, our investigation reveals that once the proposed algorithm achieves optimal decoding accuracy in both the left/right and the front/back MI tasks across participants, adding 1–5 additional ranked channels to the decoding queue does not significantly impact decoding accuracy (p>0.01, two-sample *t*-test). Additionally, when 1–7 additional ranked channels are added to the decoding queue, there is a negligible decrease in average decoding accuracy of less than 1%.

### 4.7. Latest Technologies for EEG Channel Selection and Prospects

This section describes several recent techniques for selecting channels in EEG signals and discusses their performances. Researchers have used channel selection based on the Pearson correlation coefficient combined with wavelet packet decomposition (PCC-WPD) [38] to reduce redundancy by selecting channels and features in experiments involving MI. The results showed that 124 nodes of 14 subject-specific channels out of 59 were selected, but the details regarding the criteria for selecting these channels were not provided. In the channel selection method based on the standard deviation of wavelet coefficients (stdWC) [55], researchers filtered out an average of about 4.78 MI-related channels from the original 22 channels. However, only 0–1 channel was filtered out among 44.44% of participants. Some researchers have presented an iterative channel selection graph neural network (CSGNN) for recognizing emotions [56]. This method focuses on selecting channels based on brain regions, and the results showed that the decoding accuracy decreased when the channels from different brain regions were reduced progressively by 20%. In another study on emotion recognition from EEG signals, researchers utilized an Enhanced Firefly Algorithm (EFA) [57] with brightness-distance-based attraction and a roulette-based local search strategy to select 10 important channels out of 32. However, a more detailed description of the number of selected channels and corresponding validation is required.

We calculated the ratio of redundant channels filtered by different methods to all channels in their respective data. As shown in Table 2, the proposed sparse logistic regression-based EEG channel optimization algorithm (SLRCO) is the most efficient method for removing redundant channels. Furthermore, our method is more concise than other algorithms because it does not require a sophisticated mathematical background or complex preprocessing and feature selection of the data. In addition, we thoroughly discussed the universality of the proposed channel optimization algorithm through comprehensive verification, which is a novel approach. Our research promotes the development of channel optimization algorithms for group applications. Of course, there are still some areas in our research that can be improved. For example, the core of our algorithm can be upgraded using a more robust SLR algorithm from our recent research [44]. Additionally, we could refer to existing technology’s combination method of channel and feature selection to further improve decoding accuracy or overlay brain region selection to enhance the neurophysiological significance of selected channels.

In the future development of EEG channel selection algorithms, in addition to technical improvements to the algorithm itself or combining feature selection and classification algorithms, greater attention should be paid to universality and robustness. This is of great significance for large-scale applications and reducing BCI costs.

## 5. Conclusions

This study investigated the possibility of EEG channel optimization from the individual level to the group level. First, the conventional CCS-based channel selection method was used for group analysis; satisfactory performance could not be achieved since the optimal decoding accuracy was achieved when 56 (left/right MI) or 8 (front/back MI) electrodes were selected into the decoding queue. Then, the proposed SLR-based EEG channel optimization algorithm was evaluated for individuals and for the participant group separately. The number of EEG channels for optimal decoding was only 2–16 (filtering 75–96.9% channels) with a 1.65–5.1% increase in decoding accuracy when compared with all channels. The results prove that the proposed algorithm can achieve satisfactory performance while ensuring universality when using the raw data. These results indicate that the straightforward, effective, and universal channel optimization algorithm proposed in this study is beneficial for the group application of BCI.

## Figures and Tables

**Figure 1 bioengineering-10-00664-f001:**
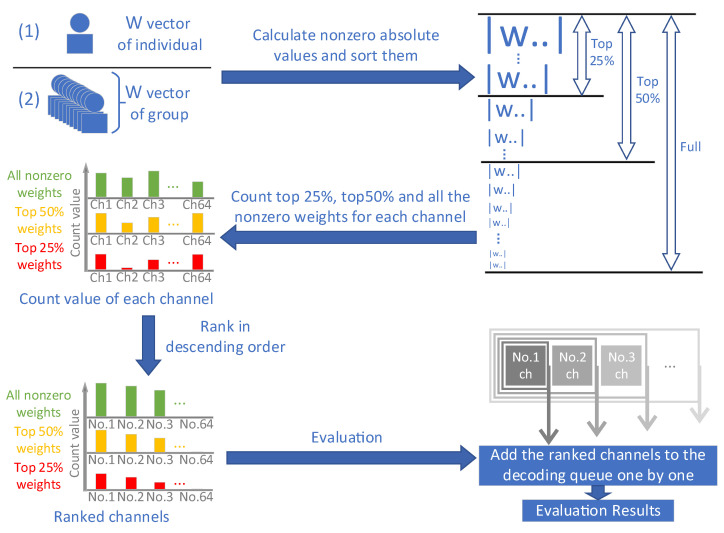
Schematic diagram of individual/group analysis for the optimization algorithm and related evaluation. In the upper left corner of the figure, it is demonstrated that (1) individual analysis involves analyzing the ‘W’ vector of each individual participant separately, while (2) the group analysis involves analyzing the ‘W’ vectors of all participants as a whole. The varying sizes of the ‘W’ character in the upper right corner represent differences in magnitude for nonzero elements within weight vectors. The count values of each channel under three different counting intervals (full/top 50%/top 25% intervals) are displayed and ranked in descending order on the lower left corner of the figure, with the horizontal axis representing channels, and the vertical axis representing count value. In addition, the lower right corner of the figure shows that the ranked channels are added one by one to a decoding queue for evaluation.

**Figure 2 bioengineering-10-00664-f002:**
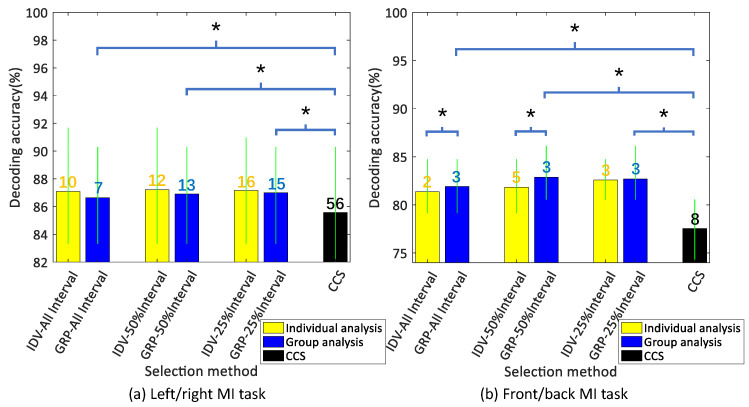
Evaluation of channel optimization performance of proposed algorithm and CCS. In the left/right and front/back MI task, the mean decoding accuracy of the optimal decoding queue of the 3 counting intervals (full/top 50%/top 25% intervals) in the individual analysis (abbreviated as ‘IDV’) and the group analysis (abbreviated as ‘GRP’) of the proposed optimization algorithm are shown in the blue and yellow bars in the chart. The group analysis results of the CCS method are represented by the black bar in the chart. The number of channels in the optimal decoding queue is marked on the bar chart. The *p*-value of a two-sample *t*-test between individual–group analysis and between the group analysis of proposed algorithm–CCS are marked with an asterisk in the figure (‘*’: p<0.01).

**Figure 3 bioengineering-10-00664-f003:**
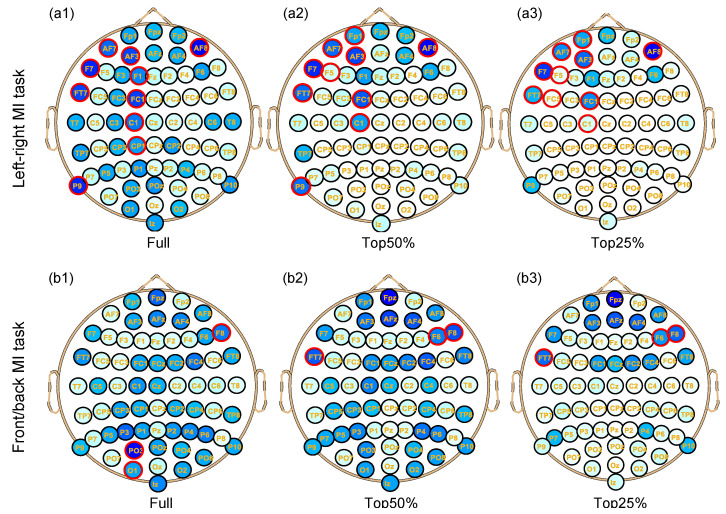
Spatial distribution of weight count values for channels in the group analysis. The upper subfigures (**a1**–**a3**) represent the spatial distribution of three counting intervals (full/top 50%/top 25% separately in the left/right MI task, while the lower subfigures (**b1**–**b3**) represent the same in the front/back MI task. The color of each electrode mark from light to dark indicates the degree of selection from low to high. The channels commonly selected among individuals are marked in red circles.

**Figure 4 bioengineering-10-00664-f004:**
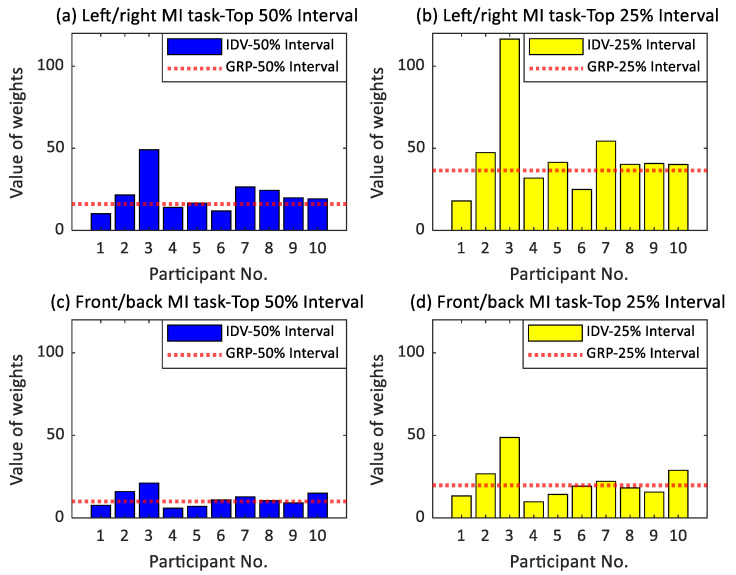
Bar chart of the absolute weight values of the lower boundary of individuals and the group. The upper subfigures (**a**,**b**) represent the left/right MI task, while the lower subfigures (**c**,**d**) represent the front/back MI task. In the group analysis (abbreviated as ‘GRP’), the red line indicates the lower boundary value of the top 50% and top 25% intervals, while for individual analysis (abbreviated as ‘IDV’), blue and yellow bars represent their respective lower boundary values of the top 50% and top 25% intervals. The abscissa shows participant serial numbers, and the ordinate displays absolute weights.

**Table 1 bioengineering-10-00664-t001:** Optimal decoding accuracy and related number of channels for group analysis and the commonly selected channels among the individuals. The table displays the decoding performance of 3 counting intervals (full/top 50%/top 25%) in both individuals and group analysis. Each string of numbers represents the average decoding accuracy, standard deviation, and number of channels in the decoding queue, respectively. Higher decoding accuracies are highlighted in bold.

Left/right MI task	Full	Top 50%	Top 25%
Group analysis	**86.63 (±4.93)**, 10	**86.91 (±5.27)**, 12	**87.00 (±5.22)**, 16
Commonly optimized channels among individuals	86.09 (±5.21), 10	86.50 (±5.17), 10	86.39 (±5.35), 10
**Front/back MI task**	**Full**	**Top 50%**	**Top 25%**
Group analysis	**81.91 (±4.28)**, 3	**81.87 (±4.12)**, 3	**82.59 (±4.11)**, 3
Commonly optimized channels among individuals	81.42 (±4.26), 3	81.14 (±4.97), 3	81.12 (±5.07), 3

**Table 2 bioengineering-10-00664-t002:** A comparison between our proposed algorithm and the latest EEG channel selection techniques that were recently published. Items that require attention are highlighted in bold.

Algorithm Name	Experiment Category	No. Participants	No. Original Channels	No. Retained Channels	Ratio of Removed Channels	Universality Discussion
PCC-WPD	MI	2	59	14	76.27%	No
stdWC	MI	9	22	17.22	21.72%	No
CSGNN	Emotion	32	32	25–26	20%	No
EFA	Emotion	32	32	22	31.25%	No
SLRCO	MI	10	64	3/10	**84.38%/95.31%**	**Yes**

## Data Availability

The data presented in this study are available on request from the corresponding author.

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
