# Peer review of "Sparse Logistic Regression-Based EEG Channel Optimization Algorithm for Improved Universality across Participants"

_bioengineering, 2023, doi:10.3390/bioengineering10060664_

Round 1

Reviewer 1 Report

The presented manuscript on the method for optimizing EEG data processing, aimed at reducing redundant information, demonstrates favorable characteristics within a formal and scientific context. The manuscript exhibits a well-organized structure and exhibits commendable readability. The introduction of the background information is thorough, accompanied by appropriate citations/.

A few minor issues:

1.     The term ‘top 50% interval’ or ‘top 25% interval’ is probably not commonly used. Please check the literature to see if there are more accurate descriptions for it.

2.     what is the method used for evaluating the decoding accuracy?  Please elaborate and give references in the introduction. 

There are some minor grammar issues, such as ‘proved’ in line 25 should be ‘proven’. Also some incorrect use of definite/indefinite articles

Author Response

Thank you very much for your review work!

Please see the attachment file.

:-)

Reviewer 2 Report

While evaluating the proposed algorithm for EEG channel optimization, I found some limitations and issues which the author need to consider to make this paper accepted 

1. The author has considered a minimal sample size for evaluating the algorithm's performance. It needs to be more extensive and more diverse population sample size.

2. The authors only compare the proposed algorithm with the conventional correlation coefficient-based method because they had yet to take some latest algorithms.

3.  The performance of the proposed algorithm is evaluated within the presence of noise, artifacts, or other interferences? Do these affect EEG signal quality? 

4. no literature review section is there? do the authors claim that no related work has been done ?? also try to add a table for comparison with state of the art which includes disadvantages and advantages 

5. add a contribution section in introduction 

6. The authors assume that the same electrode positions are used across different participants. However, EEG data acquisition may vary between participants in practice due to individual differences in head shape and size. So authors are advised to consider very carefully as the electrode placement is required for optimal performance.

7.  I suggest that authors evaluate its performance in real-world settings, which may involve more complex and noisy EEG data.

8. I have founded some grammatical errors , kindly proof read the manuscript

9.    Does your proposed algorithm work with fault tolerance .

10.  Please add future directions in details 

minor grammatical errors

Author Response

(The authors gave the same response as above.)

Round 2

Reviewer 2 Report

I accept this manuscript in current form.